# Hypoimmunogenic Human iPSCs for Repair and Regeneration in the CNS

**DOI:** 10.3390/cells14161248

**Published:** 2025-08-13

**Authors:** Haiwei Zhang, Hongxia Zhou, Xugang Xia, Qilin Cao, Ying Liu

**Affiliations:** 1Center for Translational Science, Florida International University, 11350 SW Village Pkwy, Port St. Lucie, FL 34987, USA; haizhang@fiu.edu (H.Z.); hozhou@fiu.edu (H.Z.); xxia@fiu.edu (X.X.); qicao@fiu.edu (Q.C.); 2Robert Stempel College of Public Health and Social Work, Florida International University, Miami, FL 33199, USA; 3Biomedical Science PhD Program, Herbert Wertheim College of Medicine, Florida International University, Miami, FL 33199, USA

**Keywords:** universal stem cells, transplantation, immune rejection

## Abstract

Human induced pluripotent stem cells (iPSCs) can be genetically engineered to evade host immune recognition, rendering them hypoimmunogenic and suitable as “universal donor” cells for allogeneic transplantation. Such modifications enable the development of off-the-shelf iPSC-derived therapeutics that are immediately available for clinical use without the need for patient-specific derivation or immunosuppression. This review focuses on recent developments in strategies for generating hypoimmunogenic human iPSCs, with particular emphasis on their applications in central nervous system (CNS) cell therapy and repair. We assess immunomodulatory factors based on their immune functions and potential roles in CNS development and disease, with the goal of identifying strategies to use these factors either individually, in combination, or alongside gene editing to reduce immune rejection without compromising neurogenesis or tissue repair.

## 1. Introduction

Human induced pluripotent stem cells (iPSCs) are a promising cell source for regenerative therapies aimed at restoring or replacing damaged tissues and organs. However, while the concept of using patient-specific iPSC-derived products as personalized medicine is appealing, it remains time-consuming and costly, as the cells must be derived from the patient to avoid immune reactions. One potential alternative is standardized, hypoimmunogenic iPSC-derived cells that can avoid immune rejection by the patient. These so-called “universal cells” [1,2,3,4,5] can thereby be given to any patient, bypassing the need to derive an iPSC line from every patient’s own cells. Although the viability of hypoimmunogenic iPSC derivatives continues to be debated [6,7], studies with universal cells in mice and rhesus macaques have demonstrated that they can reduce the risk of immune rejection following transplantation [8,9,10,11]. Human hypoimmunogenic iPSCs have also undergone testing in humanized animal models for various organs such as the heart, liver, and pancreas, as well as more recently in the context of the central nervous system (CNS) [11,12,13].

## 2. Strategies for Creating Universal Cells

A summary of strategies for creating hypoimmunogenic, universal cells for CNS applications can be found in Table 1.

### 2.1. Cells from Haplotype Banks of iPSCs

Creating a haplotype bank of human iPSCs is a strategic way to offer immune-compatible cell therapies to a broad population without the need for fully personalized lines or performing additional gene editing in iPSCs. The first step is to identify common human leukocyte antigen (HLA) haplotypes within the target population, typically using data from bone marrow registries or blood donor databases. The next step is to prioritize the most frequent haplotypes to achieve maximum population coverage with the fewest number of iPSC lines. For example, in Japan, researchers estimated that iPSCs derived from around 50 individuals homozygous at key HLA loci (A, B, DR) could potentially match over 90% of the population. This approach holds tremendous promise for creating scalable, widely applicable cell therapies while minimizing the risk of immune rejection. A recent phase I/II clinical trial by Takahashi’s group on iPSC-derived dopaminergic cell therapy for Parkinson’s disease [14] utilized a clinical-grade human iPSC line established from the peripheral blood of a healthy donor who was homozygous for the most common HLA haplotype in the Japanese population. This haplotype (HLA-A24:02, HLA-B52:01, HLA-C12:02, HLA-DRB115:02, HLA-DQB106:01, and HLA-DPB109:01) matches approximately 17% of the Japanese population [15]. One limitation of the haplotype bank approach is that it is best suited to populations with relatively lower genetic diversity, such as the Japanese population. In more genetically diverse populations such as those in the U.S., this method may be less effective due to the increased complexity of haplotype structures. Therefore, a more flexible approach would need to be developed to accommodate the broader spectrum of genetic variation found in these populations [16]. Another concern associated with grafting cells from a haplotype bank is the potential for immune rejection mediated by minor histocompatibility antigens (mHAgs) [17,18,19,20,21]. mHAgs are peptides derived from polymorphic proteins that differ between donor and recipient. When presented by HLA/MHC molecules on either donor or recipient cells, they can elicit T-cell responses, triggering graft rejection even in MHC-matched transplants [22]. Though traditionally considered an immune-privileged site, the CNS is not exempt from immune rejection. Mismatches at mHAgs can induce T-cell-mediated responses that lead to gradual or chronic rejection of neural grafts. Therefore, long-term monitoring is necessary for cases utilizing haplotype-matched iPSC neural grafts.

### 2.2. Gene-Edited hiPSCs

While the haplotype cell bank approach can be effective for certain populations, it is not universally applicable and requires careful planning and preparation well in advance. A more flexible strategy involves gene editing to modify the expression of HLA class I and class II molecules.

#### 2.2.1. Targeting HLA-I

HLA class I complexes are key regulators of immune surveillance, presenting intracellular peptides to cytotoxic T cells and interacting with natural killer (NK) cells. Classical class I molecules, HLA-A, -B, and -C, are highly polymorphic, widely expressed, and major targets of alloreactive T cells. Two main strategies have been developed to modulate HLA class I expression for immune evasion. One approach targets the B2M gene, which encodes β2-microglobulin. β2-microglobulin is an essential subunit required for the proper folding and surface expression of all HLA class I molecules. Without it, cells are unable to present functional HLA class I complexes on their surface. Consequently, they can avoid detection and destruction by cytotoxic T cells. However, while classical HLA class I molecules (HLA-A, -B, and -C) are the primary targets of T-cell-mediated rejection, non-classical molecules such as HLA-E and HLA-G can play key roles in inhibiting NK cell activity and promoting immune tolerance. Thus, knocking out B2M is a double-edged sword that reduces T-cell-mediated rejection but possibly increases NK-mediated rejection. To fully realize the benefits of B2M deletion, additional steps are necessary to restore or mimic the immunoregulatory functions of HLA-E and HLA-G.

To circumvent this, a second gene-editing strategy for evading HLA class I-mediated immune recognition involves selectively disrupting only the classical HLA class I genes, HLA-A, HLA-B, and HLA-C, at their respective genomic loci [9]. This approach is feasible and efficient, as HLA-B and HLA-C are located in close proximity, allowing them to be simultaneously targeted with a single CRISPR editing step. As a result, only two separate gene edits are required to eliminate the expression of all three classical HLA class I molecules [9].

HLA-E single-chain trimers (SCTs) consist of mature human β2-microglobulin, a canonical HLA-E-binding peptide, and the mature HLA-E heavy chain. Surface expression of HLA-E SCTs on porcine endothelial cells (pECs) provides protection against human NK cell-mediated cytotoxicity, representing a promising strategy to mitigate immune rejection [23]. Notably, targeted knock-in of the HLA-E trimer at the B2M genomic locus eliminates endogenous B2M expression while enabling expression of an HLA-E/β2M fusion protein. This approach permits inducible surface expression of HLA-E while abolishing expression of classical HLA class I molecules (HLA-A, -B, and -C). As a result, engineered human embryonic stem cells (ESCs) and their differentiated derivatives evade recognition by both allogeneic cytotoxic T cells and NK cells [24].

#### 2.2.2. Targeting HLA-II

Disrupting the function of HLA class II molecules is relatively straightforward. The classical HLA class II molecules HLA-DR, -DQ, and -DP are heterodimers consisting of an α and a β chain encoded by distinct loci. In addition to the classical class II molecules, several non-classical HLA class II molecules, such as HLA-DM and HLA-DO, play specialized regulatory roles. Both classical and non-classical HLA class II molecules contribute to immune surveillance, autoimmunity, and transplant rejection. To eliminate HLA class II expression, a widely adopted strategy is the knockout of CIITA (class II transactivator), a master regulatory gene essential for the transcription of all HLA class II genes. Disruption of CIITA effectively silences HLA class II expression, even under inflammatory conditions such as interferon gamma (IFNγ) stimulation, which would normally induce robust upregulation [8,9]. As an alternative approach, targeting components of the RFX transcriptional complex, such as RFXANK, RFX5, or RFXAP, can also suppress HLA class II expression. These proteins are critical for the assembly and function of the RFX complex, which binds to the promoter regions of HLA class II genes and drives their transcription. Disruption of any of these components impairs promoter activation, thereby preventing class II expression at the transcriptional level [1,5].

#### 2.2.3. Impact of HLA Editing on Differentiation, Function, and Integration of Neural Lineage Grafts

Properly HLA-edited hypoimmunogenic iPSC lines by CRISPR maintain normal properties when rigorous genomic quality controls are applied. Clones with undesired on-target or off-target mutations are typically discarded during the screening process. This section will discuss whether HLA editing (e.g., B2M-null) will affect neural differentiation, maturation, circuit formation, or synaptogenesis of hypoimmunogenic iPSC-derived neural lineage cells. As a critical component of HLA-I molecules, B2M plays critical roles in neural differentiation, maturation, and cognitive function across various neurological conditions. Its effects are highly context-dependent, involving both HLA-I-dependent and -independent mechanisms. In amyotrophic lateral sclerosis (ALS), B2M expression is upregulated in motor neurons and appears to support neuronal plasticity, potentially through interactions with synaptic receptors such as PirB. B2M knockout in ALS mouse models accelerates disease progression, suggesting a beneficial role for B2M in maintaining neuromuscular connectivity [25]. In contrast, B2M has been reported to suppress neurogenesis and neural plasticity in aging and neurodegenerative diseases while supporting neuronal survival in other contexts [26]. Recent work on hypoimmunogenic cells containing B2M knockout do not report any defects in neural differentiation or graft integration [12,13]. Due to limited number of reports on hypoimmunogenic neural grafts so far, further studies are needed to determine whether HLA editing negatively affects the differentiation, function, and integration neural lineage grafts.

### 2.3. Overexpression of Immunomodulatory Factors, Their Role in Immune Modulation, and Impact on Neural Cell Function

A third approach to generate hypoimmunogenic cells is the overexpression of multiple immunomodulatory factors [11,27]. For example, Harding et al. overexpressed eight transgenes—PD-L1, CD200, CD47, H2-M3, FasL, SerpinB9, CCL21, and MFGE8—in mouse embryonic stem cells (mESCs), enabling grafts derived from these “cloaked” mESCs to persist long term in fully immunocompetent, allogeneic hosts. Consistently, the human counterparts of these genes, when expressed in hESCs, suppressed the activation and proinflammatory responses of allogeneic human peripheral blood mononuclear cells (PBMCs) in vitro [11,27]. Recently, our lab combined HLA editing with the immunomodulatory factor MIF to generate hypoimmunogenic hiPSCs and tested their derived NPCs in SCI animal models [13]. Immunomodulatory factors must meet two key criteria: first, they should cooperatively inhibit immune rejection alongside other factors or gene-editing strategies; second, they must not interfere with neural differentiation. In this section, we discuss the immune functions of these factors and their potential roles in neural development and evaluate whether their individual or combined use is better for CNS transplantation applications (Figure 1 and Table 2).

#### 2.3.1. PD-L1

Programmed death-ligand 1 (PD-L1), encoded by the CD274 gene, is an immune checkpoint protein that regulates T-cell activity by binding to PD-1. The interaction of PD-L1/PD-1 suppresses T-cell responses and helps to maintain immune homeostasis. PD-L1 is often overexpressed in various tumors as a mechanism to evade immune surveillance. Hence, PD-L1 has become a key therapeutic target in immunotherapy for several types of cancer [29,30,31,32]. In the CNS, as part of immune regulation, PD-L1 expressed on astrocytes interacts with PD-1 on microglia to suppress the release of the pro-inflammatory cytokines IL-1β and TNF-α, thereby attenuating autoimmune inflammation within the CNS. PD-L1 also modulates the peripheral immune response, shifting CD4 + T-cell polarization toward anti-inflammatory phenotypes, including regulatory T cells (Tregs) and Th2 cells [33].

PD-L1 also plays a critical role in neural development through both immune modulation and direct interactions with neural cells. For instance, aberrant PD-L1 activity disrupts oligodendrocyte progenitor cell (OPC) association with blood vessels, mediated by astrocyte-secreted CSRP1, which subsequently impairs oligodendrocyte maturation and compromises white matter integrity during early brain development [34]. Additionally, PD-L1 promotes astrocytic end-foot coverage of blood vessels and maintaining proper astrocyte morphology, an essential feature for supporting neurovascular integrity. Furthermore, PD-L1 contributes to neuroprotection by enhancing the integrity of the blood–brain barrier (BBB) following injury by upregulating the expression of tight junction proteins, including claudin-5 and ZO-1 [35]. PD-L1 has also been shown to potently inhibit the excitability of human nociceptive neurons and act as a pain inhibitor and neuromodulator [36].

#### 2.3.2. SERPINB9

SERPINB9 is a serine protease inhibitor that plays a crucial role in protecting cells from apoptosis by directly inhibiting Granzyme B, a potent cytotoxic enzyme released by cytotoxic T cells and NK cells. By blocking Granzyme B-mediated cleavage of intracellular substrates, SERPINB9 serves as a molecular safeguard against immune cell-induced cytotoxicity [37,38,39]. Interestingly, SERPINB9 is only detected in the human cerebellum, but not in other regions of the CNS (https://www.proteinatlas.org/ENSG00000170542-SERPINB9, accessed on 30 June 2025). Nevertheless, SERPINB9 has been shown to protect somatic and stem cells, including neural progenitor cells, from Granzyme B-induced cell death in peripheral and in vitro models. This data supports the hypothesis that SERPINB9 may serve a similar cytoprotective function during neurogenesis, potentially ensuring the survival of developing neural cells in the face of immune-mediated stress or inflammation.

#### 2.3.3. MFGE8

Milk fat globule-EGF factor 8 (MFGE8), also known as lactadherin, is a secreted glycoprotein. One of the key immune functions of MFGE8 is the promotion of apoptotic cell clearance. Acting as a bridging molecule, MFGE8 binds phosphatidylserine on apoptotic cells and integrins (αvβ3/αvβ5) on phagocytes, facilitating the efficient engulfment of dying cells and reducing the release of pro-inflammatory cellular contents and maintaining immune homeostasis [40,41,42]. MFGE8 exerts anti-inflammatory effects by downregulating the production or inhibiting the release of pro-inflammatory molecules including TNF-α, IL-1β, IL-6, and high-mobility group box 1 (HMGB1), in part through inhibition of key signaling pathways like MAPK and NF-κB [43]. Additionally, MFGE8 directly inhibits inflammasome-induced IL-1β production in macrophages via the β3 integrin and P2X7 receptor pathways, thus limiting excessive innate immune activation [44]. MFGE8 also promotes the polarization of macrophages toward an M2 anti-inflammatory phenotype and regulates the uptake of necrotic cells in dendritic cells [45,46,47]. MFGE8 preserves quiescent neural stem cells by inhibiting the mTORC1 signaling pathway [48,49]. Hence, MFGE8 is an anti-inflammatory factor with potential roles in supporting the preservation of neural stem cells.

#### 2.3.4. CD200

CD200 is a key immunomodulatory glycoprotein expressed on neurons that regulates microglial activity and synaptic refinement during neural development. Through its interaction with the CD200 receptor (CD200R) on microglia, CD200 maintains microglia in a homeostatic state. This suppressive interaction is crucial for preventing excessive neuroinflammation during critical periods of brain development. In the absence of CD200, microglia exhibit a hyperactive phenotype, characterized by increased phagocytic activity and increased pro-inflammatory signaling which disrupts the normal refinement of neural circuits [50,51,52]. Beyond inflammation control, CD200 plays a protective role in synaptic maintenance. It functions as a “do-not-eat me” signal to microglia, safeguarding active synapses from unnecessary or excessive pruning [53,54]. CD200 also promotes neurogenesis and neuronal survival. In vitro studies show that exogenous CD200 enhances the survival of neurons in primary culture and facilitates neurogenesis [51]. The CD200 and CD200R axis is dynamically regulated, and dysregulation has been implicated in synaptic loss associated with neurodegenerative diseases [55]. Hence, CD200 is a critical regulator of neuroimmune interactions, synaptic preservation, neural development, and neurological disorders.

#### 2.3.5. CD47

CD47 plays a pivotal role in neural development. It regulates synaptic pruning and promotes neurite outgrowth. By binding to SIRPα on microglia [56], CD47 acts as a “do-not-eat-me” signal, inhibiting microglial-mediated phagocytosis of synaptic elements. Conversely, neural activity dynamically modulates CD47 expression on synaptic membranes; decreased neural activity leads to a reduction in synaptic CD47 levels, thereby enhancing susceptibility to microglial pruning [57,58]. In contrast, CD47 overexpression is associated with brain overgrowth and neurodevelopmental delays, likely due to disruption of normal synaptic and progenitor cell pruning [59]. Additionally, CD47 overexpression also resists killing by NK cells, mediated through CD47 on target cells and SIRPα on NK cells [60,61].

#### 2.3.6. IL-17

Interleukin-17 (IL-17) can be produced by a variety of immune cells, including Th17 cells, γδ T cells, and innate lymphoid cells. It coordinates both innate and adaptive immune responses and exerts complex immunomodulatory functions that are critical for maintaining immune homeostasis and defending against pathogens. IL-17 also synergizes with IFNγ to potentiate macrophage-mediated nitric oxide production and supports epithelial barrier function by upregulating tight junction proteins and activating resident stem cells to promote tissue repair at sites of injury [62]. Despite its protective roles, IL-17 is a key driver of inflammatory pathology in autoimmune and chronic inflammatory diseases. Overexpression of IL-17 can lead to excessive neutrophilic inflammation, tissue destruction, and bone resorption, mediated by upregulation of pro-osteoclastogenic factors RANKL and tissue-degrading enzymes matrix metalloproteinases (MMPs) [63,64]. IL-17 also perpetuates chronic inflammation by activating the NF-κB and MAPK pathways, leading to sustained expression of TNF, IL-6, CCL20, and CXCL10 [65,66,67]. Additionally, IL-17 regulates Th17/Treg balance. IL-17-producing Tregs show diminished suppressive function, potentially exacerbating inflammation [68]. IL-17 also aids in the recruitment of Th1 cells to sites of infection, thereby facilitating IFNγ-driven pathogen clearance [67,69].

IL-17 regulates neurogenesis, synaptic plasticity, and neuroimmune interactions. Under physiological conditions in the adult hippocampus, endogenous IL-17 suppresses neurogenesis. IL-17 knockout mice show increased proliferation and survival of adult-born neurons in the dentate gyrus, enhanced synaptic transmission, and elevated expression of proneuronal genes in neural progenitor cells [70]. Conversely, exogenous IL-17 has been shown to stimulate neuronal differentiation in human iPSC-derived NPCs via ERK1/2 and mTORC1 signaling [71]. Additionally, IL-17 produced by γδ T cells or Th17 cells interacts with microglia and astrocytes, shaping neuroinflammatory environments that influence neurodevelopment [72,73]. Overall, IL-17 acts as both a regulator of neurogenesis and a mediator of neuroimmune crosstalk.

#### 2.3.7. IL-10

Interleukin-10 (IL-10) is a multifunctional cytokine that plays a central role in immune regulation and neural development and disease. A summary of potential roles of IL-10 in hypoimmunogenic cells is shown in Figure 2.

One of the most prominent roles of IL-10 is its anti-inflammatory and immunosuppressive function. IL-10 limits inflammation by several mechanisms. First, it suppresses the production of key pro-inflammatory cytokines such as TNF-α, IL-1, IL-6, IL-12, and IFN-γ in macrophages, dendritic cells, and T cells. Second, IL-10 reduces antigen presentation capacity by downregulating the expression of MHC class II molecules, co-stimulatory molecules such as CD80 and CD86, and adhesion molecules like ICAM-1 on antigen-presenting cells (APCs), thereby impairing effective T-cell activation [74,75]. Third, it impairs microbicidal mechanisms within macrophages by inhibiting the production of nitric oxide and reactive oxygen species [74]. Finally, IL-10 promotes T-cell anergy by blocking CD28 co-stimulatory signaling, leading to long-term inactivation of T cells [74]. Despite its anti-inflammatory reputation, IL-10 also exhibits paradoxical immunostimulatory and pro-survival effects in specific cellular contexts. In B cells, IL-10 enhances survival, supports proliferation, and promotes differentiation into antibody-producing plasma cells [74]. It also plays a pivotal role in the development and function of Tregs, supporting their immunosuppressive activity and helping to maintain immune tolerance [76]. In NK cells, IL-10 enhances proliferation, migration, and cytotoxic activity, enabling these cells to efficiently target infected or transformed cells. In mast cells, IL-10 increases cell survival, upregulates FcεRI (high-affinity IgE receptor) expression, and promotes the release of pro-inflammatory cytokines. In certain conditions, IL-10 can even stimulate CD8 + T cells, enhancing their production of cytotoxic molecules like Granzyme B, boosting MHC expression, and increasing their ability to kill target cells.

The dual roles of IL-10 are particularly evident in disease pathogenesis, where its functions can be protective or detrimental depending on the context. During infections, IL-10 is essential for controlling harmful inflammation, such as in sepsis, where unchecked immune activation can lead to organ failure and death. However, by suppressing immune responses, IL-10 may also hinder effective pathogen clearance and promote persistent infection [74]. IL-10 modulates oxidative phosphorylation pathways to regulate decidual immune cell function during pregnancy, helping to maintain fetal tolerance and maternal immune balance [77]. In the context of the CNS, IL-10 enhances neuronal survival through multiple mechanisms. First, it inhibits apoptosis by activating the Jak-Stat3 and PI3K/Akt signaling, upregulating anti-apoptotic proteins Bcl-2 and Bcl-xL [74,78]. Second, it mitigates excitotoxicity by suppressing the production of inflammatory cytokines such as TNF-α and IL-1β [78,79]. Third, it reduces oxidative stress in neurons by modulating mitochondrial function and enhancing mitophagy through regulation of the Ddit4/mTORC1 signaling axis, promoting cellular homeostasis and survival under metabolic stress [78,80,81,82,83]. IL-10 also suppresses excessive differentiation of NPCs into neurons, thus maintaining a balanced progenitor pool during development [78,84]. IL-10 additionally modulates immune responses within developing neural tissues through several pathways. It influences microglial polarization by promoting the transition to an M2 anti-inflammatory phenotype. These M2-polarized microglia and macrophages produce lower levels of pro-inflammatory cytokines such as TNF-α, IL-1β, and IL-6, fostering a neuroprotective and regenerative environment conducive to normal neural development [78,85]. IL-10 also regulates astrocyte activity by limiting reactive astrogliosis and enhancing the expression of TGF-β, a cytokine that supports tissue repair and anti-inflammatory responses [74,85]. In addition, IL-10 promotes the induction of regulatory T cells in the CNS, which are instrumental in suppressing trauma-induced autoimmunity and help to ensure immune tolerance [78]. IL-10 has been shown to promote neurogenesis and synaptogenesis and reduce neuroinflammation and glial scar formation [84]. In spinal cord injury models, exogenous delivery of IL-10 has been shown to promote axon regeneration and remyelination. These effects are primarily mediated through activation of key signaling cascades, including the STAT3 and PI3K/Akt/mTOR pathways, which facilitate repair processes and enhance neuronal resilience [74,78]. To this effect, disruptions in IL-10 signaling have been linked with neurodevelopmental disorders. For instance, aberrant IL-10 activity is associated with altered emotional behavior and cognitive impairments, implicating it in the proper maturation of neural circuits and stress responses during development [74,86]. However, the effects of IL-10 vary across experimental contexts. In vitro studies often demonstrate enhanced neuronal survival and reduced inflammation, whereas in vivo models may reveal less favorable outcomes, including impaired clearance of neurotoxic substances, highlighting the species- and environment-specific nature of IL-10 [74]. Despite this complexity, IL-10 remains a strong immunomodulatory factor for hypoimmunogenic cell engineering.

#### 2.3.8. MIF

Macrophage Migration Inhibitory Factor (MIF) is a pleiotropic cytokine secreted by a wide range of cell types, including T cells, macrophages, and glial cells. It plays a dynamic and context-dependent role in both innate and adaptive immune responses. For instance, MIF modulates macrophage polarization by shifting macrophages toward an anti-inflammatory phenotype that supports tissue repair and immune regulation. MIF can downregulate the expression of NKG2D, an NK cell-activating receptor. Through transcriptional repression of NKG2D, MIF reduces NK cell activation and cytotoxicity and facilitates immune evasion, an effect that may be particularly beneficial in promoting graft tolerance following cell transplantation. At the molecular level, MIF functions as a homotrimer and signals primarily through CD74, a cell surface receptor that activates downstream MAPK signaling cascades [87]. This pathway is implicated in regulating inflammatory responses, promoting cell survival, and mediating broader immune suppression. A recent study showed that MIF binds to CD37, triggering phosphorylation at the Y13 residue to significantly reduce phagocytic activity and allowing cells to avoid immune clearance [87]. Beyond its immunomodulatory properties, MIF also plays a critical role in neural development and repair. It is endogenously expressed in multiple regions of the developing brain and spinal cord, where it contributes to neuroprotection, promotes neuronal differentiation, and enhances neurite outgrowth. Additionally, MIF has been shown to exert a protective effect in ALS by reducing the accumulation of misfolded and aggregated SOD1 protein [88]. Loss-of-function studies have demonstrated that MIF deficiency leads to reduced expression of key neuronal markers such as PSA-NCAM and doublecortin, and pharmacological inhibition of MIF impairs neuronal proliferation and neurite extension, particularly in the hippocampus. Hence, MIF plays an essential role in the maturation and integration of neural cells, which is particularly relevant in the context of regenerative therapies. MIF has therefore emerged as a promising dual-function candidate capable of modulating host immune responses while also supporting neural differentiation and graft survival. Our recent work further supports this notion, showing that MIF significantly reduces NK cell-mediated cytotoxicity without compromising the survival or neural differentiation of transplanted cells in SCI animal models [13]. Collectively, these data support MIF as a valuable tool in the development of hypoimmunogenic neural lineage cell therapies.

#### 2.3.9. HLA-G

HLA-G is a non-classical, immune-modulatory class Ib antigen [89,90,91,92] that orchestrates tolerance in both innate and adaptive immunity. Membrane-bound and soluble HLA-G binds to the inhibitory receptors ILT2, ILT4, and KIR2DL4 to directly suppress NK cell cytotoxicity, diminish T- and B-cell activation, and impair dendritic cell maturation [93]. HLA-G also diminishes immune responses by promoting regulatory T-cell development and upregulating other non-classical class I molecules (e.g., HLA-E). It has been reported that HLA-G is essential for maternal–fetal tolerance, preventing immune attack on the fetus, limiting excessive inflammation, and maintaining peripheral immune homeostasis. Recent reports support that HLA-G can function as an immune checkpoint protein [94].

### 2.4. Advantages and Disadvantages of Strategies for Generating Hypoimmunogenic iPSCs

As discussed above, three primary strategies have been explored for reducing immune rejection in hiPSC-derived NPCs and other neural lineage cells: (a) engineering cells to express specific immunomodulatory factors; (b) gene editing to delete key HLA molecules; (c) employing allogeneic cell transplantation alongside immunosuppressive therapies. Each approach presents unique advantages and limitations, and the optimal choice depends on clinical objectives, safety profiles, scalability, and regulatory considerations (Table 3).

The first approach involves engineering hiPSC-derived neural cells to express immunoregulatory genes. Factors such as PD-L1, MFGE8, and CD47 can locally modulate immune responses and promote immune tolerance without the need for systemic immunosuppression. This is particularly advantageous in immune-privileged environments like the CNS, where low HLA-I expression already provides partial protection from immune attack. Additionally, multiple immunomodulatory molecules can be combined to tailor immune evasion to specific transplantation contexts or patient genetic profiles. However, the expression of engineered genes can be unstable or silenced after transplantation, reducing the efficacy of immune modulation. Moreover, the long-term effects of immune factor overexpression remain uncertain, with potential risks including unintended changes in cell proliferation, differentiation, or integration affected by the overexpressed immunomodulatory factors. Clinical experience with this strategy is also limited compared to more established methods.

A second strategy focuses on the HLA gene editing system to reduce immune recognition. By deleting or selectively modifying HLA molecules, the goal is to create universal donor cells suitable for allogenic transplantation to unmatching recipients. This method reduces the presentation of alloantigens and diminishes T-cell-mediated rejection. This strategy also offers scalability, as a small number of hiPSC lines could serve a large, diverse patient population. However, a complete loss of HLA molecules may provoke strong responses from NK cells, which detect the absence of the non-classical HLA-E and HLA-G as a signal for cell destruction. To mitigate this, these inhibitory HLAs like HLA-E or HLA-G could be selectively kept. However, weakening immune surveillance can carry tumorigenic risks, particularly if any cells become transformed. Even with successful HLA editing, residual immunological recognition via minor antigens can still trigger rejection, and the clinical application of this method continues to face ethical and regulatory challenges.

The third approach, traditional allogeneic cell transplantation combined with immunosuppressive therapy, is widely established in solid organ and hematopoietic transplants and has been applied in some neural transplantation scenarios. However, this method can increase the likelihood of complications such as severe infections, metabolic complications, kidney toxicity, and malignancies. Furthermore, this approach is still susceptible to the threat of graft-versus-host disease (GvHD), especially when there is imperfect HLA matching. While certain areas of the CNS offer immune privilege, others do not, and even in protected regions, immune responses can arise over time or after trauma. The financial and logistical burden of long-term patient monitoring and management also adds to the challenges of this approach.

No single strategy is ideal in all clinical contexts. Autologous iPSC-derived neural cells, derived from a patient’s own tissues, offer the safest option in terms of immune compatibility, but it is time-consuming and expensive to generate them individually for each patient. HLA-edited or immunoregulatory gene-engineered hypoimmunogenic cells offer scalable, off-the-shelf solutions, particularly as gene editing and our understanding of immune evasion continue to improve. Meanwhile, conventional allogeneic transplantation with immunosuppression remains the most practical in the short term, especially when gene editing is not feasible or regulatory conditions are restrictive, though it requires acceptance of long-term health risks for the patient. Emerging evidence supports a hybrid approach: combining partial HLA editing, such as retaining inhibitory HLAs to avoid NK cell attack, with short-term, low-dose immunosuppression to facilitate early graft survival. This compromise offers a promising balance between safety, efficacy, and clinical practicality.

In conclusion, HLA editing, when used alongside targeted immune cloaking strategies, represents the most promising direction for future hypoimmunogenic neural cell transplantation. It holds the potential to create universal donor cells with minimized rejection risk and without the need for lifelong immunosuppressive therapy. However, key technical, regulatory, and safety challenges must still be overcome before this strategy can be broadly implemented in clinical settings. In the meantime, treatment strategies should continue to be individualized based on patient needs and clinical conditions while ongoing research continues to optimize and refine these promising therapeutic approaches.

## 3. Generating Hypoimmunogenic Stem Cells Specifically for Clinical Applications in the CNS

When creating hypoimmunogenic cells, it is essential to keep clinical translation as the ultimate goal. Critical considerations include the potential for off-target effects of CRISPR/Cas9 gene editing, which may introduce unintended mutations at genomic sites with partial sequence homology to the target. Equally important are unexpected on-target effects, including large deletions, insertions, and chromosomal translocations [95,96]. Both can disrupt gene function and compromise genomic stability, raising safety concerns for therapeutic use. To mitigate these risks, the editing strategy should be simplified and the overall process streamlined. Rigorous quality control, including high-resolution whole-genome sequencing, is essential to detect and minimize undesired alterations to the genome. Additionally, for successful clinical application, all components must comply with current Good Manufacturing Practice (cGMP) standards. Fortunately, cGMP-grade Cas9 protein, plasmids, and iPSCs are either commercially available or can be produced under cGMP conditions.

There are limited reports on the application of hypoimmunogenic cells in the context of CNS. For example, oligodendrocyte progenitor cells derived from a hypoimmunogenic hiPSC line engineered through knockout of B2M and CIITA alone, without further modification, were successfully used to restore myelin in a Canavan disease animal model, a condition characterized by severe demyelination [12]. On the other hand, a more recent report shows that hypoimmunogenic human embryonic stem cells, generated by overexpressing eight immunomodulatory genes, can be differentiated into ventral midbrain dopaminergic neurons and transplanted into Parkinson’s disease animal models [11]. Our lab has employed a combinatorial strategy to generate hypoimmunogenic iPSCs and their neural derivatives for transplantation studies in spinal cord injury (SCI) animal models, including T-cell-deficient nude rats and humanized mice [13]. These studies collectively support the potential of hypoimmunogenic iPSCs for CNS applications. However, several challenges remain in both preclinical and clinical settings, as discussed in this section.

### 3.1. How Can Effective Integration of Hypoimmunogenic iPSC-Derived Neural Cells with Host Neural Tissue Be Achieved?

The CNS imposes distinct and stringent requirements on iPSC-derived neural grafts, distinguishing them from cell therapies intended for other tissues. Successful CNS integration necessitates not only the survival of transplanted neurons or glial cells but also their capacity to structurally and functionally incorporate into the host neural circuitry. This includes the formation of appropriate synaptic connections and active engagement with existing neural networks. Grafts that remain anatomically present but functionally isolated are unlikely to contribute meaningfully to neural repair. As such, further investigation is warranted to address the unique immunological and functional challenges associated with the application of hypoimmunogenic human iPSC-derived grafts in the CNS.

The development of hypoimmunogenic human iPSC-derived neural grafts marks a significant step forward in addressing immune barriers while enhancing functional integration within the neural circuits of the brain and spinal cord. This dual strategy of both reducing immune rejection and promoting synaptic connectivity holds the potential to benefit a broader range of neurological disorders. By attenuating host-versus-graft immune responses, these engineered grafts provide a promising platform to expand the clinical utility of iPSC-based therapies for neural repair and regeneration.

Although the brain and spinal cord have historically been considered immune-privileged sites, they harbor a variety of immune cell types such as microglia that actively participate in immune and inflammatory responses. Moreover, long-term immunosuppression is typically required for CNS xenografts in animal models, underscoring the presence of significant immune reactivity and rejection within the CNS. The therapeutic success of NPC transplantation in SCI and other neurological diseases depends not only on graft survival, but also on the appropriate differentiation of these cells into region- and disease-specific neuronal and glial subtypes. Furthermore, the formation of appropriate synaptic connections between transplanted grafts and the host neural network is a critical factor for therapeutic success. Establishing functional synapses and neural projections is essential for effective neuronal communication and coordination, which underpins meaningful functional recovery. Several important questions remain unanswered. First, it is unclear whether the use of hypoimmunogenic hiPSCs will influence the differentiation potential and integration capacity of neural grafts within the CNS. Second, how hypoimmunogenic hiPSC-NPCs will survive and differentiate in the hostile, injury-associated environment post-SCI or other neurological diseases, remains to be determined. Third, although MHC modification or immune cloaking strategies may enable these cells to evade immune rejection, it is unknown whether such evasion will support long-term graft survival in the CNS. Finally, whether bypassing host immune surveillance with hypoimmunogenic hiPSC-derived NPCs increases the risk of uncontrolled cell proliferation or tumor formation is an area of active concern. Our recent work has begun to address these critical knowledge gaps by providing foundational data that may help to accelerate the safe and effective translation of hiPSC-based therapies for neural repair and regeneration in the CNS [13]. More in vivo studies are needed to address these critical concerns.

### 3.2. Does Evasion of Host Immune Surveillance Increase the Risk of Tumor Formation and Graft Overgrowth, and How Can This Risk Be Mitigated?

An important safety concern surrounding the clinical application of hypoimmunogenic human iPSC-derived cells is their potential to evade immune surveillance and, in turn, form tumors. While the ability to avoid host immune rejection is a key advantage of hypoimmunogenic cells, this same feature raises the risk that any undifferentiated iPSCs inadvertently included in the graft could survive and proliferate in the host. Undifferentiated iPSCs have tumorigenic potential and are known to form teratomas in immunocompromised environments if not properly eliminated during the differentiation process. In the context of hypoimmunogenic cells, the risk is further amplified, as these cells are specifically engineered to evade immune detection, making it even less likely that the host immune system would recognize and remove them. Hence, high purity of the grafted neural lineage cells is critical.

This challenge underscores the critical need for stringent quality control measures prior to transplantation. Reliable methods must be developed to monitor any undifferentiated cells within the graft, as well as sensitive in vivo imaging or biomarker-based surveillance tools to detect early signs of graft-derived tumor formation post-transplantation. Equally important is the establishment of fail-safe mechanisms, such as engineered suicide genes or targeted elimination strategies, that can be activated to selectively destroy problematic cells if tumorigenesis is detected. Addressing these safety concerns is essential for advancing hypoimmunogenic iPSC-derived therapies toward safe and effective clinical use. Ultimately, clinical trials using neural derivatives such as NPCs, neuronal progenitor cells, or glial progenitor cells derived from hypoimmunogenic iPSCs will be necessary. These trials will generate essential preclinical and clinical safety data and efficacy data, with the goal of enabling grafting into any patient without the risk of immune mismatch between donor cells and recipients. Ideally, much like a conventional prescription drug, a rigorously tested hypoimmunogenic universal cell—one that has passed extensive quality control standards—could be made available off-the-shelf, ready for use at any time for any patient in need.

### 3.3. Technical Considerations for the Simultaneous Overexpression of Multiple Immunomodulatory Factors

Recent reports of eight-factor overexpression have not documented transgene silencing. However, it remains to be determined whether this stability persists over the long term following grafting. Critical factors such as deciding between safe harbor loci or random integration and the method used for transgene delivery must be carefully considered and optimized to ensure sustained expression and functional safety.

Another technical consideration involves the potential challenges of testing hypoimmunogenic hiPSC-NPCs in the CNS. This starts with the selection of an appropriate model. Theoretically, hypoimmunogenic cells of animals or non-human primates (NHPs) can be tested in corresponding species. Human hypoimmunogenic cells will often be first tested in immunocompromised animal models, then in humanized mice with or without corresponding disease models, which represent an allograft environment. The grafted NPCs may take more than three months to differentiate into mature neurons and glial cells [97]. Carefully analyzing the differentiation and maturation of grafted NPCs in long-term is a must. Commonly used humanized animal models include humanized hu-CD34+ NSG mice. While they offer several advantages, it is important to consider the limitations associated with the use of humanized mice in transplantation research. First, the immune repertoire in humanized mice may not accurately reflect the diversity and functionality of human immune cells, potentially limiting the interpretation and relevance of transplantation studies. Second, the engraftment of human cells in humanized mice can vary between individual mice, resulting in heterogeneity within the experimental population, which leads to inconsistent or unpredictable outcomes, making it challenging to interpret data. Third, mice and humans have inherent species-specific differences in cellular signaling, tissue microenvironments, and metabolic pathways. Consequently, certain cellular responses or physiological processes may not accurately mimic human physiology. Lastly, humanized mice can develop xenogeneic GvHD, which can compromise the well-being of the mice and confound experimental outcomes.

NHPs are indispensable in the translational pipeline for developing and validating hiPSC-derived hypoimmunogenic neural cell therapies. Their close genetic, anatomical, and physiological resemblance to humans makes them uniquely suited for testing the survival, function, and immune response of hypoimmunogenic cells under clinically relevant conditions, which provides a predictive platform that bridges preclinical findings with human outcomes. NHPs offer a level of translational fidelity that is critical for evaluating the engraftment, integration, and long-term survival of neural grafts. Their brain structure, immune landscape, and behavioral repertoire allow for multidimensional assessments, including motor and cognitive testing, neuroimaging, electrophysiology, and immunophenotyping, that more closely approximate what would occur in a human clinical trial. Importantly, some immune-mediated graft rejection responses are primate-specific and do not emerge in rodent models, underscoring the need for NHP validation before clinical application. Although studies specifically testing hypoimmunogenic hiPSCs in NHPs have not yet been reported, transplantation of hiPSC-derived cells in NHP models has been documented and provides a valuable foundation for evaluating universal donor cells in a clinically relevant setting [98,99,100].

## 4. Summary and Future Directions

Immune rejection is a crucial factor that is often overlooked but plays a critical role in cell transplantation therapy for CNS injury and disease. Effective modification of the immunogenicity of transplanted cells, taking into consideration the unique anatomy, physiology, and function of the CNS, is a critical consideration in cell therapy for SCI. By addressing the challenges posed by immune rejection upon grafting, the hypoimmunogenic hiPSC-NPC approach aims to accelerate the translation of allogeneic stem cell therapy and enhance the viability and success of cell transplantation therapies, ultimately improving outcomes for CNS injury patients. Ongoing research has demonstrated increased survival and the ability to avoid immune responses with hypoimmunogenic and immune modulated iPSC-NPCs [13]. In addition, the use of an FDA-approved cell line increases the potential for clinical application.

There are several important directions in which future studies on this topic should focus. First, in vivo investigations will be essential to evaluate how hypoimmunogenic cells survive, differentiate, and integrate within the CNS following transplantation. Such studies will provide insights into the functional potential of the grafts, their capacity to adopt region-specific neural phenotypes, and their ability to establish synaptic connections or modulate the local cellular milieu. Longitudinal in vivo imaging and histological analyses will help to determine how these cells behave over time within the complex CNS environment. Second, while overexpression of individual immunomodulatory genes has shown promise in enhancing immune evasion, the combinatorial effects of multiple transgenes remain largely unexplored. Future work should therefore also investigate whether overexpression of multiple immunoregulatory factors (e.g., PD-L1, CD47) leads to synergistic immune protection or introduces unintended interference. Dissecting the molecular and cellular consequences of these interactions will be critical for optimizing gene-engineered cell therapies. Third, long-term safety remains a paramount concern. Future studies should assess whether hypoimmunogenic modifications increase the risk of uncontrolled proliferation, tumor formation, or abnormal cell behavior after transplantation into the CNS. These evaluations should include extended post-transplantation monitoring in relevant animal models and the use of sensitive detection methods to uncover any potential adverse outcomes. Fourth, as these approaches advance toward clinical application, several translational issues must be addressed. Regulatory challenges, scalability of manufacturing under GMP conditions, and the reproducibility of hypoimmunogenic phenotypes across diverse genetic backgrounds also represent critical barriers that need to be resolved to ensure safe and effective clinical translation. Finally, other future areas of interest include evaluating the immunological responses in humanized animal models, developing non-invasive imaging strategies for tracking graft survival in real time, and exploring the compatibility of hypoimmunogenic strategies with other regenerative platforms, such as organoids or scaffold-based delivery systems. Furthermore, ethical and regulatory frameworks surrounding the use of genetically modified cells in the CNS will need to be continually updated in line with technological progress.

## Figures and Tables

**Figure 1 cells-14-01248-f001:**
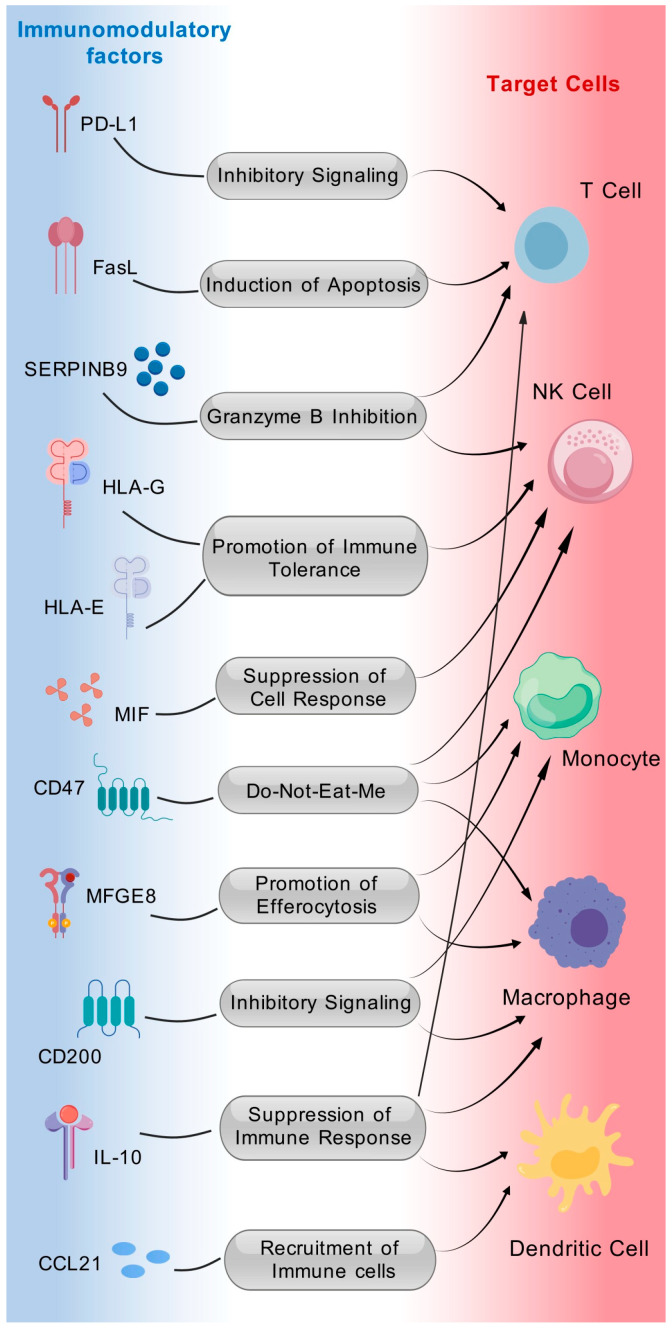
Immunomodulatory factors and their roles in generating hypoimmunogenic iPSCs. Created with BioGDP.com [28].

**Figure 2 cells-14-01248-f002:**
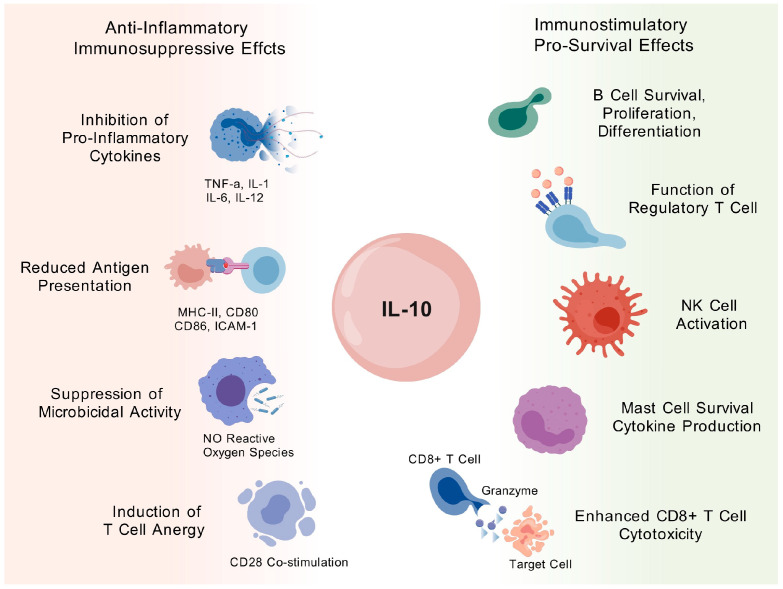
A summary of potential roles of IL-10 in hypoimmunogenic cells. Created with BioGDP.com [28].

**Table 1 cells-14-01248-t001:** Summary of strategies for generating hypoimmunogenic human iPSCs for CNS applications.

Method	Mechanism	Applications	Advantages	Disadvantages	Refs
HLA-haplotype iPSC banks	Derive iPSCs from donors homozygous for common HLA haplotypes	Parkinson’s disease (dopaminergic neurons)	Avoids gene editing; scalable for certain populations	Limited HLA coverage in diverse populations; needs large banks	[14,15]
Gene-edited hiPSCs	Knockout of B2M (HLA-I) and CIITA (HLA-II) or selective disruption of HLA-A/B/C loci	Oligodendrocyte progenitor cells in Canavan disease	Flexible; broadly applicable; off-the-shelf potential	Risk of NK cell activation if non-classical HLA-E/G not retained; CRISPR editing on-target mutation and off-target effects	[12]
Overexpression of immunomodulatory factors (PD-L1, CD200, CD47, SERPINB9, MFGE8, etc.)	Express multiple immunosuppressive genes to inhibit T cells, NK cells, macrophages	Ventral midbrain dopaminergic neurons (Parkinson’s disease)	Survival in fully immunocompetent hosts; broad immune evasion	Technical complexity; stability of transgene expression; tumor risk	[11]
Combined gene-editing and overexpression of immunomodulatory factors	Knockout of B2M (HLA-I) and CIITA (HLA-II) and overexpress MIF	Neural progenitors in SCI models	Survival and integration of neural progenitors in SCI models	Long-term evaluations are needed; risks for mutations from CRISPR editing	[13]

**Table 2 cells-14-01248-t002:** Immunomodulatory factors for generating hypoimmunogenic hiPSCs and hESCs.

Factor	Cell Type	Mechanism	Reference
PD-L1	T cells	Regulates T-cell activity by binding to PD-1, suppresses T-cell responses, maintains immune homeostasis, modulates peripheral immune response, shifts CD4 + T-cell polarization toward anti-inflammatory phenotypes	[11,27]
FASL	T cells	Induces apoptosis in target cells expressing Fas receptor	
SERPINB9	T cells, NK cells	Directly inhibits Granzyme B, protecting cells from apoptosis	[11,27]
MFGE8	Monocytes, Macrophages	Promotes apoptotic cell clearance, dampens excessive inflammatory responses, promotes immune tolerance	[11,27]
CD200	Monocytes, Macrophages	Regulates microglial activity, maintains them in a homeostatic, surveillance state, protective role in synaptic maintenance	[11,27]
CD47	Monocytes, Macrophages	Acts as a “do-not-eat-me” signal, inhibiting microglial-mediated phagocytosis, protects killing from NK cells	[8,9,10,11,27]
CCL21	Dendritic cells	Chemokine that attracts immune cells	[11,27]
IL-10	Immune-suppressive	Suppresses pro-inflammatory cytokines, reduces antigen presentation, impairs microbicidal mechanisms, promotes T-cell anergy, enhances B-cell survival/proliferation/differentiation, supports Tregs	
HLA-G	NK cells	Inhibits NK cell activity, promotes immune tolerance	[11,27]
HLA-E	NK cells	Inhibits NK cell activity, promotes immune tolerance	[23,24]
MIF	NK cells	Suppresses NK cell response	[13]

**Table 3 cells-14-01248-t003:** Comparison of approaches for minimizing immune rejection in stem cell-derived neural cell transplantation.

Strategy	Description	Advantages	Limitations
Overexpression of immunomodulatory factors	hiPSC-derived NSCs or neural cells are modified to express immunomodulatory factors (PD-L1, MFGE8, and CD47, etc.) to achieve immune cloaking	Localized immune modulation without systemic immunosuppression Particularly effective in immune-privileged sites like the CNS Allows flexible combinations of immunoregulatory genes tailored to patient or transplantation context	Transgene expression may be unstable or silenced post-transplantation Long-term effects of overexpression on cell proliferation, differentiation, or integration remain uncertain Limited clinical precedent compared to other strategies
HLA gene editing	Targeted deletion or modification of HLA class I and/or class II genes to reduce immune recognition and create universal donor cells	Reduces T-cell-mediated rejection by lowering alloantigen presentation Fewer hiPSC lines needed to treat large populations	Requires retention or reintroduction of inhibitory HLAs (e.g., HLA-E, HLA-G) to suppress NK activation Weakening immune surveillance may increase tumorigenic risk Residual immune responses (e.g., minor antigens) can still occur
Allogeneic transplantation with immunosuppression	Use of allogeneic donor-derived neural cells with matched HLA typing in combination with pharmacological immunosuppression	Does not require complex genetic engineering	Requires lifelong immunosuppression Risks of infection, cancer, metabolic issues, and nephrotoxicity Risk of GvHD Immune privilege of CNS may not fully protect against rejection Long-term patient monitoring is costly and complex

## Data Availability

No new data were created or analyzed in this study. Data sharing is not applicable to this article.

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
