# Peer review of "Hypoimmunogenic Human iPSCs for Repair and Regeneration in the CNS"

_cells, 2025, doi:10.3390/cells14161248_

Round 1

Reviewer 1 Report

Comments and Suggestions for Authors

The manuscript discusses multiple immunomodulatory factors; however, it doesn’t provide a critical assessment of the advantages and disadvantages of using these factors vs HLA-editing or allogeneic cell transplantation followed by immunosuppression.  The preferred strategy for generating hypoimmunogenic neural cells for transplantation is not suggested/discussed as well.

On lines 715-717, the authors emphasize the off-target effect as a chief risk factor. However, this risk can be mitigated by employing editing procedures with minimal off-target effects and/or by selecting clones that lack these effects. It seems that the potential interference of primary genetic edits with differentiation, integration, and function of neuronal cells is a more critical issue than just CRISPR/Cas9 edits.

Authors should acknowledge the role of nonhuman primates in the proper assessment of the function of engrafted cells and the potential of using mismatched cells, along with immunosuppression. The important papers discussing autologous vs allogeneic neural cell transplantation could be discussed (DOI: 10.1038/s41591-021-01257-1, DOI: 10.1016/j.brainresbull.2025.111297).

Although the authors discussed hypoimmunogenic cells in relation to neural cell transplantation, they should mention several important papers concerning non-neural cells, such as hypoimmunogenic studies with islet cells and https://doi.org/10.1038/s41587-023-01784-x, endothelial cells doi: 10.1038/s41587-022-01426-8, and the potential for using unmodified cells along with immunosuppression https://doi.org/10.1038/s41586-024-08463-0.

Hypoimmunogenic approaches using HLA-E should be discussed (doi: 10.1038/nbt.3860, doi: 10.1111/j.1399-3089.2007.00378.x.)

Subtitles are very simplistic and don’t reflect the discussed topics. For example, 2.3 discusses immunomodulatory factors, their role in immune modulation, and their impact on neural cell function, which could limit their applicability for generating neurons and glia for transplantation purposes.     However, the title is “overexpression of immunomodulatory factors”.

To be consistent, the impact of HLA editing on differentiation, function and integration should be discussed in Chapter 2.2.

There is an inconsistency between the factors presented in Fig. 1 and Table 2. Not all factors listed in Table 2 are shown in Figure 1. Several factors such as H2-M3, HLA-G and MIF are not discussed et all. Critical mechanisms related to action of several factors in Table 2 are not listed. For example, CD47 overexpression also protects from killing by NK cells. This should be added to the Table 2 and discussed.

Lines 560-561 “Reports showing knocking out B2M could affect the neural differentiation.” should provide references.

Comments on the Quality of English Language

Additional work is required to improve language and grammar.

Author Response

The manuscript discusses multiple immunomodulatory factors; however, it doesn’t provide a critical assessment of the advantages and disadvantages of using these factors vs HLA-editing or allogeneic cell transplantation followed by immunosuppression. The preferred strategy for generating hypoimmunogenic neural cells for transplantation is not suggested/discussed as well.

Thank the reviewer for the valuable suggestion. We have included a new subsection (Subsection 2.4. Advantages and disadvantages of strategies for generating hypoimmunogenic iPSCs) and a new table (Table 3. Comparison of approaches for minimizing immune rejection in stem cell-derived neural cell transplantation). The Subsection and Table are located at Lines 410-477.

On lines 715-717, the authors emphasize the off-target effect as a chief risk factor. However, this risk can be mitigated by employing editing procedures with minimal off-target effects and/or by selecting clones that lack these effects. It seems that the potential interference of primary genetic edits with differentiation, integration, and function of neuronal cells is a more critical issue than just CRISPR/Cas9 edits.

Thank the reviewer for pointing out. We completely agree with the reviewer and have added a new subsection (Subsection 2.2.3. Impact of HLA editing on differentiation, function and integration of neural lineage grafts) to discuss the potential effects of gene editing on grafts differentiation and function. The subsection can be found at Lines 138-158.

Authors should acknowledge the role of nonhuman primates in the proper assessment of the function of engrafted cells and the potential of using mismatched cells, along with immunosuppression. The important papers discussing autologous vs allogeneic neural cell transplantation could be discussed (DOI: 10.1038/s41591-021-01257-1, DOI: 10.1016/j.brainresbull.2025.111297).

Thank the reviewer for the valuable advice. We have added a new paragraph focusing on the importance of using nonhuman primate models and the relevant references. This can be found at Lines 606-621.

Although the authors discussed hypoimmunogenic cells in relation to neural cell transplantation, they should mention several important papers concerning non-neural cells, such as hypoimmunogenic studies with islet cells and https://doi.org/10.1038/s41587-023-01784-x, endothelial cells doi: 10.1038/s41587-022-01426-8, and the potential for using unmodified cells along with immunosuppression https://doi.org/10.1038/s41586-024-08463-0.

Thank the reviewer for the valuable advice. These references have been added. Specifically,

Ref. 10: Hu et al, 2023. https://doi.org/10.1038/s41587-023-01784-x;

Ref. 8: Deuse et al, 2019, was included in the original submission. doi: 10.1038/s41587-022-01426-8;

Ref. 99. Jebran et al, 2025. https://doi.org/10.1038/s41586-024-08463-0.

Hypoimmunogenic approaches using HLA-E should be discussed (doi: 10.1038/nbt.3860, doi: 10.1111/j.1399-3089.2007.00378.x.)

Thank the reviewer for the valuable advice. We have added the references (refs.23, 24) and a new paragraph discussing this elegant strategy for creating hypoimmunogenic cells. This can be found at Subsection 2.2.1, Lines 109-118.

Subtitles are very simplistic and don’t reflect the discussed topics. For example, 2.3 discusses immunomodulatory factors, their role in immune modulation, and their impact on neural cell function, which could limit their applicability for generating neurons and glia for transplantation purposes.     However, the title is “overexpression of immunomodulatory factors”.

Thank the reviewer for the suggestion. We have revised subtitles to reflect the content. Subsection 2.3. Overexpression of immunomodulatory factors, their role in immune modulation, and impact on neural cell function.

To be consistent, the impact of HLA editing on differentiation, function and integration should be discussed in Chapter 2.2.

Thank the reviewer for the suggestion. We have added Subsection 2.2.3 to discuss the Impact of HLA editing on differentiation, function and integration of neural lineage grafts.

There is an inconsistency between the factors presented in Fig. 1 and Table 2. Not all factors listed in Table 2 are shown in Figure 1. Several factors such as H2-M3, HLA-G and MIF are not discussed et all. Critical mechanisms related to action of several factors in Table 2 are not listed. For example, CD47 overexpression also protects from killing by NK cells. This should be added to the Table 2 and discussed.

Thank the reviewer for pointing out. We have remade Figure 1 to be consistent with Table 2. We have also revised Table 2 to include critical mechanisms for CD47.

We have also added subsections 2.3.8 and 2.3.9 to discuss HLA-G and MIF.

Because H2-M3 is a mouse MHC class Ib molecule and was one of the immunomodulatory factors used to generate hypoimmunogenic mouse iPSCs, while our manuscript focuses on human hypoimmunogenic cells, we have removed details on H2-M3 in the revised version.

Lines 560-561 “Reports showing knocking out B2M could affect the neural differentiation.” should provide references.

Thank the reviewer for pointing out. We have re-organized this part to Subsection 2.2.3. The reference (Staats, et al, 2013) has been added. The discussion can found at Lines 138-158.

Comments on the Quality of English Language

Additional work is required to improve language and grammar.

Thank the reviewer for the suggestion. We have asked a scientist who is a native English speaker to edit the language in the revised manuscript.

Reviewer 2 Report

Comments and Suggestions for Authors

The author discusses human induced pluripotent stem cells (iPSCs) that can be genetically engineered to evade host immune recognition, making them hypoimmunogenic and suitable as "universal donor" cells for allogeneic transplantation. The review focuses particularly on their application in central nervous system (CNS) cell therapy and repair.
Overall, this is a thorough and well-balanced review, supported by a broad and relevant selection of references.
However, I have a few comments and suggestions for improvement:
•    Section 2.1: "Cells from haplotype banks of iPSCs"
The author should address the risk of immune rejection related to minor histocompatibility antigens (mHAgs), which remains a concern even with haplotype-compatible iPSC lines. This point is well discussed in the following review and could be incorporated: https://pubmed.ncbi.nlm.nih.gov/25262921/
•    Section 2.3: Immunomodulatory factors
While this section is somewhat long, it effectively highlights the dual role of many immunomodulatory factors in immune regulation and neural development. I recommend adding a brief introductory note at the beginning of section 2.3 (just before 2.3.1) to underscore this critical issue, especially given its relevance in CNS-specific PSC-based therapies.
•    Lines 530–542
There is a repetition about CRISPR unintended off-targets (pointed at two different places in the paragraph): simplify. Additionally, the author should include a reference to potential on-target mutations, which are equally important and can impact gene function or cell behavior. For example, see: https://www.ncbi.nlm.nih.gov/pubmed/30059492
•    Line 543
The sentence "For the CNS, limited reports have been shown for the use of hypoimmunogenic cells in the CNS." contains a redundancy. I suggest rephrasing to avoid repeating "CNS." For instance:
"There are limited reports on the application of hypoimmunogenic cells in the context of the CNS."

Author Response

The author discusses human induced pluripotent stem cells (iPSCs) that can be genetically engineered to evade host immune recognition, making them hypoimmunogenic and suitable as "universal donor" cells for allogeneic transplantation. The review focuses particularly on their application in central nervous system (CNS) cell therapy and repair.
Overall, this is a thorough and well-balanced review, supported by a broad and relevant selection of references.
However, I have a few comments and suggestions for improvement:

  •    Section 2.1: "Cells from haplotype banks of iPSCs"
    The author should address the risk of immune rejection related to minor histocompatibility antigens (mHAgs), which remains a concern even with haplotype-compatible iPSC lines. This point is well discussed in the following review and could be incorporated: https://pubmed.ncbi.nlm.nih.gov/25262921/

Thank the reviewer for the valuable suggestion. We have added a new portion and references discussing on mHAgs, which are essential for immune rejection for allografts. This can be found at Lines 70-78.

  •    Section 2.3: Immunomodulatory factors
    While this section is somewhat long, it effectively highlights the dual role of many immunomodulatory factors in immune regulation and neural development. I recommend adding a brief introductory note at the beginning of section 2.3 (just before 2.3.1) to underscore this critical issue, especially given its relevance in CNS-specific PSC-based therapies.

Thank the reviewer for the valuable advice. We have added an introductory note to briefly summarize the immunomodulatory factors. This can be found at Lines 170-175.

  •    Lines 530–542
    There is a repetition about CRISPR unintended off-targets (pointed at two different places in the paragraph): simplify. Additionally, the author should include a reference to potential on-target mutations, which are equally important and can impact gene function or cell behavior. For example, see: https://www.ncbi.nlm.nih.gov/pubmed/30059492

Thank the reviewer for pointing out. We have revised this paragraph and added discussion on on-target mutations with references (Lines 481-493)

  •    Line 543
    The sentence "For the CNS, limited reports have been shown for the use of hypoimmunogenic cells in the CNS." contains a redundancy. I suggest rephrasing to avoid repeating "CNS." For instance:
    "There are limited reports on the application of hypoimmunogenic cells in the context of the CNS."

Thank the reviewer for the suggestion. We have revised the sentence as advised (Lines 494-495).

Round 2

Reviewer 1 Report

Comments and Suggestions for Authors

Manuscript significntly improved. All my criticisms are properly addressed.